# Impact of urbanisation on the gaps of hypertension prevalence, awareness and treatment among older age in China: a cross-sectional study

Qiutong Yu [1,2] Genyong Zuo [1,2]

[1]Centre for Health Management and Policy Research, School of Public Health, Cheeloo College of Medicine, Shandong University, Jinan, Shandong, China
[2]NHC Key Laboratory of Health Economics and Policy Research, Shandong University, Jinan, Shandong, China

**Correspondence to**
Dr Genyong Zuo;
smartyong@sdu.edu.cn

## ABSTRACT

**Objectives** To examine the impact of urbanisation on the prevalence, awareness and treatment of hypertension among elderly in China.
**Design** This cross-sectional study used data from the most recent nationally representative Chinese Longitudinal Healthy Longevity Survey, 2018.
**Setting** People in urban and rural communities from 500 sample areas in 22 Chinese provinces.
**Participants** After exclusion, this study surveyed 9859 participants in the final analysis.
**Primary and secondary outcome measures** The main dependent variables were prevalence, awareness and treatment of hypertension defined as (1) systolic blood pressure (BP)≥140 mm Hg, diastolic BP≥90 mm Hg or (2) taking antihypertensive drugs. Hypertension awareness was defined as a previous diagnosis of hypertension by a health professional, and hypertension treatment was defined as undergoing BP treatment.
**Results** The prevalence of hypertension was lower among semiurbanised adults than among non-urbanised rural adults (OR=0.94, 95% CI=0.90 to 0.99; p<0.05). The probabilities of awareness (OR=1.10, 95% CI=1.01 to 1.20; p<0.05) and treatment (OR=1.17, 95% CI=1.08 to 1.26; p<0.001) of hypertension were significantly lower among non-urbanised adults than among urban-born adults. Urbanisation in eastern (OR=0.93, 95% CI=0.88 to 0.99; p<0.05) and western China (OR=1.11, 95% CI=1.01 to 1.23; p<0.05) was associated with the prevalence of hypertension. The urbanisation level was also associated with hypertension awareness and treatment in eastern (OR=1.17, 95% CI=1.04 to 1.32; p<0.01; OR=1.26, 95% CI=1.14 to 1.40; p<0.001), central (OR=1.31, 95% CI=1.05 to 1.63; p<0.05; OR=1.29, 95% CI=1.08 to 1.55; p<0.01) and western China (OR=1.28, 95% CI=1.07 to 1.53; p<0.01; OR=1.34, 95% CI=1.15 to 1.57; p<0.001). The Blinder-Oaxaca decomposition suggested that approximately 42% and 39% of the urban–rural gap in hypertension awareness and treatment, respectively, could be attributed to coefficient difference.
**Conclusions** Public health programmes and policies for chronic diseases should adjust with urbanisation and combine individual-centred strategies.

## STRENGTHS AND LIMITATIONS OF THIS STUDY

⇒ Our large sample from 22 provinces in China can clearly reveal the attribution difference of hypertension between urban and rural communities.
⇒ Combine hukou (China's unique household registration system), current residence and birthplace to comprehensively show the level of urbanisation.
⇒ Blinder-Oaxaca decomposition is used to determine the extent of gap in prevalence, awareness and treatment of hypertension between urban and rural.
⇒ The major limitation of our study is that we could not make causal inferences because of the cross-sectional nature.

Approximately 7.3 million people worldwide annually die from hypertension.[2] Between 1975 and 2015, the number of hypertensive patients increased from 594 million to more than 1.1 billion, mostly in low-income and middle-income countries (LMICs).[3] Hypertension is a silent killer that rarely manifests symptoms in its early stages. However, elevated blood pressure (BP) increases the risk of chronic diseases, and early awareness and treatment are key to reducing the risk.[4] Yet, the proportion of people who were diagnosed and treated in LMICs was reportedly lower than that of high-income countries.[5]

The Grand Challenges emphasised the need to study the impact of urbanisation on hypertension.[6] One of the key drivers of urbanisation in LMICs was rural-to-urban migration, which was accompanied by a sharp increase in the prevalence of chronic diseases due to socioeconomic development, changes in lifestyles and inequalities in healthcare utilisation between urban and rural areas.[7] Therefore, it is meaningful to assess the impact of rural-to-urban migration on hypertension owing to its huge public health consequences.

This is particularly true in China, which has experienced a period of rapid urbanisation.

## INTRODUCTION

Hypertension is a major risk factor of stroke, heart disease and renal failure.[1]

The seventh national census report in 2020 recorded 375.18 million individuals, an increase of 69.75% over 2010.[8] Moreover, the national cross-sectional survey from 2012 to 2015 showed that the prevalence of hypertension in the population aged>60 years was 53.2%.[9] Due to the imbalance between urban and rural development as well as the different levels of urbanisation development in eastern, northeastern, central and western China,[10 11] there are gaps in the awareness and treatment of hypertension. The Chinese government has made great efforts to fill these gaps by explicitly investing in human resources to improve the number of doctors in township health centres and other public health programmes.[12] However, rural/urban disparities concerning health services still exist,[13] and the urban–rural gap among the eastern, northeastern, central and western regions is gradually widening.[14]

Previous studies have assessed the relationship between urbanisation and the prevalence, awareness and treatment of hypertension according to socioeconomic status.[7 13 15] The results of these analyses have highlighted that education, income and the proportion of insured urban adults were significantly higher than those of rural adults, and these factors were significantly positively correlated with the awareness and treatment of hypertension. Some studies have also estimated the temporal trend of the prevalence, awareness and treatment of hypertension with the development of urbanisation.[16 17] Furthermore, there is a strong body of evidence for an increased risk of hypertension in children and adolescents in eastern, central and western China, showing differing socioeconomic profiles.[14] Studying the relationship between the prevalence, awareness, and treatment among hypertension and urbanisation, and making comparisons within different areas of urbanisation development can provide insight into the development of rural and urban areas and their respective impacts on hypertension.

In view of current research, available reports and publications have largely examined and investigated hypertension and urbanisation, but the relationship between urbanisation and the prevalence, awareness and treatment of hypertension in China has not been fully studied. First, most previous studies were based only on hukou (China's unique HRS) or current residence and did not show the characteristics of the floating population.[18] Second, in recent years, there have been few studies on the awareness and treatment of urbanisation and hypertension in the context of different economic regions throughout China.[19] Third, although a study has used the Blinder-Oaxaca decomposition to explain urban–rural disparities in hypertension,[16] most studies did not accurately quantify the extent to which the gaps in prevalence, awareness and treatment of hypertension can be explained by urban–rural differences. Therefore, in this study, the urbanisation classification was used to show in detail the changes in sociodemographic characteristics caused by population mobility, and the Blinder-Oaxaca classification was used to evaluate the gaps in prevalence,

awareness and treatment of hypertension caused by urban and rural areas.

The main purpose of this study was to examine (1) whether the level of urbanisation is associated with the prevalence, awareness and treatment of hypertension in eastern, northeastern, central and western China after controlling for education and income and (2) whether there is a gap in prevalence, awareness and treatment of hypertension between urban-born and non-urbanised rural adults, and (3) the extent to which any gaps in prevalence, awareness and treatment of hypertension can be explained by urban–rural differences. To the best of our knowledge, this is the first study to comprehensively show the impact of rural-to-urban migration on the prevalence, awareness and treatment of hypertension based on the sociodemographic characteristics of the floating population in China. We hypothesise that the research results will provide insights for countries undergoing rapid social transformation and urbanisation to develop effective measures to reduce health disparities and achieve healthy ageing.

## METHODS
### Sampling
In this study, we used secondary data from the 2018 Chinese Longitudinal Healthy Longevity Survey (CLHLS), a longitudinal population-based study of people aged>65 years in China. These data were the most recently available and therefore best fits China's current population situation.[20] Using a multistage stratified proportional probability sampling design, approximately 16 000 older people in urban and rural communities were randomly selected from 500 sample areas in 22 provinces. In total, 9859 individuals were found eligible for this study, excluding 493 participants who refused to answer, 103 participants with technical problems, 870 participants who answered 'not applicable' and 4549 participants with missing data (figure 1).

### Variables
The main dependent variables in this study were prevalence, awareness and treatment. We defined hypertension as (1) systolic BP≥140 mm Hg, diastolic BP≥90 mm Hg or (2) taking antihypertensive drugs.[21] After the study participant rested for at least 5 min, a research assistant measured the BP of the right arm two times using a mercury sphygmomanometer at 1 min intervals, and the mean of the two measurements was calculated. For bedridden participants, BP was measured in the reclining position. Hypertension awareness was defined as respondents knowing that they had been previously diagnosed with hypertension by a health professional, assessed by the question, 'Have you been hospital diagnosed with hypertension?' Hypertension treatment was defined as taking BP treatment, correlating with the question, 'Whether you take antihypertensive drugs?'

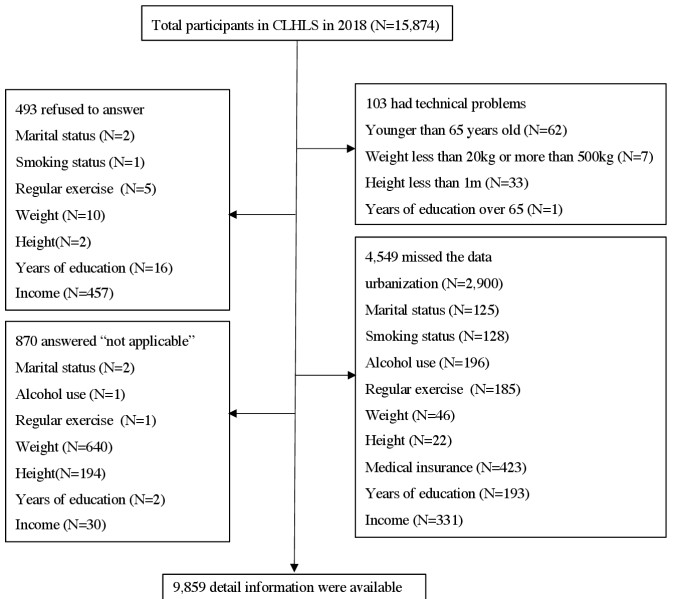

Total participants in CLHLS in 2018 (N=15,874)

493 refused to answer
Marital status (N=2)
Smoking status (N=1)
Regular exercise (N=5)
Weight (N=10)
Height(N=2)
Years of education (N=16)
Income (N=457)

103 had technical problems
Younger than 65 years old (N=62)
Weight less than 20kg or more than 500kg (N=7)
Height less than 1m (N=33)
Years of education over 65 (N=1)

870 answered "not applicable"
Marital status (N=2)
Alcohol use (N=1)
Regular exercise (N=1)
Weight (N=640)
Height(N=194)
Years of education (N=2)
Income (N=30)

4,549 missed the data
urbanization (N=2,900)
Marital status (N=125)
Smoking status (N=128)
Alcohol use (N=196)
Regular exercise (N=185)
Weight (N=46)
Height (N=22)
Medical insurance (N=423)
Years of education (N=193)
Income (N=331)

9,859 detail information were available

**Figure 1** Study flowchart of participants selection (aged 65 years or over) from Chinese Longitudinal Healthy Longevity Survey 2018 survey data.

The key independent variable was the urbanisation type. Four types of urbanisation were identified: (1) non-urbanised: individuals were born and lived in rural areas, and held the rural hukou all the time; (2) semiurbanised: individuals were born in rural areas but had moved into and lived in urban areas while still holding the rural hukou; (3) fully urbanised: individuals were born in rural areas, had moved in and lived in urban areas and obtained the urban hukou; and (4) urban born: individuals were born and lived in urban areas and held the urban hukou all the time.[22]

We adjusted for demographic characteristics, including age (elderly age (60–75 years), senile age (75–90 years), long livers (>90 years)),[23] sex (male, female), marital status (married: married and living together; not married: widowed, divorced, separated or never married) and geographical region. We divided the regions by their economic belt: eastern China (including Beijing, Tianjin, Hebei, Shanghai, Jiangsu, Zhejiang, Fujian, Shandong, Guangdong and Hainan provinces); central China (including Shanxi, Anhui, Henan, Hubei and Hunan provinces); western China (including Guangxi, Chongqing, Sichuan and Shaanxi provinces); and northeast China (including Liaoning, Jilin and Heilongjiang provinces).

Socioeconomic characteristics included education, income, body mass index (BMI) and medical insurance. Education and income were continuous variables representing years of schooling and total household income in the previous year, respectively. BMI was calculated as weight in kilograms divided by height in metres squared and classified as overweight and obese ($\geq 24$ kg/m$^2$), or others (<24 kg/m$^2$).[24] Medical insurance comprised urban resident basic medical insurance (URBMI), urban employee basic medical insurance (UEBMI), new rural cooperative medical scheme insurance (NCMS) and others such as commercial medical insurance and government-free medical insurance.

Lifestyle behaviour included smoking status, alcohol drinking and exercise. Participants who answered 'smoking now' were recognised as yes, as was alcohol use and regular exercise.

## Statistical analysis

Data were analysed using STATA V.15.0 for descriptive analysis, logistic regression and the Blinder-Oaxaca decomposition. Descriptive statistical data on hypertension prevalence, awareness and treatment among adults of the four types of urbanisation are presented proportionally, and the corresponding $\chi^2$ and p values were used to check whether there were statistically significantly differences among adults in the four urbanisation levels. Continuous variables in demographic, socioeconomic and lifestyle behaviours were estimated by means of averages and categorical variables by proportions. The $\chi^2$ test was used for dichotomous variables, and the t-test was used for continuous variables.

We used logistic regression to examine the differences among adults of the four types of urbanisation in a multivariate analysis adjusted for confounding variables. Adjusted ORs and p values are reported. The model is specified as follows:

$$\text{logit}\,(P_i)\,\text{In}\left(\frac{P_i}{1-P_i}\right) \\ = \beta_0 + \beta_1\,\text{urbanization}_i + \sum_1^n a_n X_{ni} \tag{1}$$

where $P_i$ represents the probability of the prevalence, awareness and treatment of hypertension. Urbanization$_i$ represents the four types of urbanisation: non-urbanised, semiurbanised, fully urbanised and urban born. $\beta_0$ is a constant term, the coefficients $\beta_1$ represent disparities among the four types of urbanisation and $X_{ni}$ are control variables. In addition, subgroup analyses were performed using a logistic model to examine the impact of urbanisation on the prevalence, awareness and treatment of hypertension stratified by region in China after controlling for education and income.

The Blinder-Oaxaca decomposition is a commonly used disparities decomposition method. Based on the above-mentioned logit regression, this study will further use the Blinder-Oaxaca decomposition to decompose the disparities in the prevalence, awareness and treatment of hypertension between urban and rural areas into three parts: endowment effect, coefficient effect and interaction effect (the effect caused by the endowment effect and the coefficient effect at the same time).[25 26]

The factors that affected the prevalence, awareness and treatment functions of hypertension were estimated for the two sets of samples, and the regression coefficients of the estimated equation were obtained. The Blinder-Oaxaca decomposition of the differences between the two sets was calculated as follows:

$$\bar{y}_a - \bar{y}_b = \quad (\bar{x}_a - \bar{x}_b) + \beta_b + \bar{x}_b (\beta_a - \beta_b)$$
$$+ (\bar{x}_a - \bar{x}_b)(\beta_a - \beta_b) \tag{2}$$

Where $a$ were the adults living in urban areas and $b$ were the adults living in rural areas; $(\bar{x}_a - \bar{x}_b) \beta_b$ was the 'endowments effect' (or 'explained component'); and $\bar{x}_b (\beta_a - \beta_b)$ was the 'coefficients effect' (or 'the unexplained component'). $(\bar{x}_a - \bar{x}_b)(\beta_a - \beta_b)$ can be explained by the interaction between the endowment effect and coefficient effect.

### Patient and public involvement

All data in this study were derived from the CLHLS dataset.

## RESULTS
### Descriptive results

The background characteristics of the participants are listed in table 1. The prevalence, awareness and treatment of hypertension in rural-born adults were significantly lower than those in urban-born adults (p<0.001). The descriptive results showed that non-urbanised, semiurbanised, fully urbanised and urban-born adults also differed significantly in all control variables (p<0.001), and years of education increased with the level of urbanisation. The income of people born in cities was more than two times that of those who were non-urbanised and semiurbanised. The BMI of people born in cities was the highest, and it increased with the level of urbanisation. Rural-born adults participating in the NCMS accounted for approximately 70.2% of the sample.

### Association of urbanisation and the prevalence, awareness and medical use of hypertension

We regressed the three binary variables (prevalence, awareness and treatment) on demographic characteristics, socioeconomic characteristics and lifestyle behaviour, as shown in table 2. After controlling for confounding variables, semiurbanised adults had lower odds of hypertension than did non-urbanised adults (OR=0.94, 95% CI=0.90 to 0.99; p<0.05). Urban-born adults were 10% (OR=1.10, 95% CI=1.01 to 1.20; p<0.05) more likely to be diagnosed with hypertension than the reference group (rural born). There was a similar difference in treatment; urban-born adults were 17% (OR=1.17, 95% CI=1.08 to 1.26; p<0.001) more likely to be medicated than non-urbanised adults.

The risk of hypertension was 12% (OR=1.12, 95% CI=1.06 to 1.20; p<0.001) more likely in the senile age group than in the reference group (elderly age). Long livers were less likely to be diagnosed (OR=0.84, 95% CI=0.77 to 0.91; p<0.001) or medicated (OR=0.85, 95% CI=0.79 to 0.91; p<0.001) for hypertension than were elderly patients. The association between geographical region and hypertension was significantly different. Specifically, participants living in central and western

China had 89% (OR=0.89, 95% CI=0.86 to 0.92; p<0.001) and 93% (OR=0.93, 95% CI=0.91 to 0.96; p<0.001) lower odds of hypertension than participants in eastern China, respectively. Adults with UEBMI/URBMI had a higher probability of awareness (OR=1.67, 95% CI=1.25 to 2.28; p<0.01) and treatment (OR=1.40, 95% CI=1.08 to 1.82; p<0.05).

### Subgroup analysis

Table 3 lists the results of the stratified analyses. Regarding geographic location, the prevalence of hypertension decreased by a factor of 0.93 for each level of urbanisation in eastern China (OR=0.93, 95% CI=0.88 to 0.99; p<0.05). Additionally, the relationship between urbanisation and the prevalence of hypertension was significant in western China, and participants were 1.11 times more likely to develop hypertension for each level of urbanisation (OR=1.11, 95% CI=1.01 to 1.23; p<0.05). No significant association was noted between urbanisation and the prevalence of hypertension in northeastern or central China. The urbanisation level was associated with hypertension awareness and treatment in eastern China (OR=1.17, 95% CI=1.04 to 1.32; p<0.01; OR=1.26, 95% CI=1.14 to 1.40; p<0.001), central China (OR=1.31, 95% CI=1.05 to 1.63; p<0.05; OR=1.29, 95% CI=1.08 to 1.55; p<0.01) and western China (OR=1.28, 95% CI=1.07 to 1.53; p<0.01; OR=1.34, 95% CI=1.15 to 1.57; p<0.001). No significant associations were noted between urbanisation and hypertension awareness and treatment in northeast China.

### Decomposition analysis

The decomposition results in table 4 show that the probabilities of being hypertensive were 61.18% (95% CI=0.60 to 0.62, p<0.001) for urban adults and 61.11% (95% CI=0.60 to 0.63, p<0.001) for rural adults. Both endowment and coefficient effects were significant. The results for awareness and treatment were somewhat different. The overall urban–rural differences in the predicted probabilities of awareness and treatment were 6.68% (95% CI=0.04 to 0.09, p<0.001) and 10.54% (95% CI=0.08 to 0.13, p<0.001), respectively. Both endowment and coefficient effects were significant in the logistic decompositions for awareness and treatment, although the former was more important. Approximately 2.82% of the total urban–rural difference in hypertension awareness could be attributed to the coefficient effect. The treatment findings were comparable. The total differences and differences attributed to the coefficient effect between urban and rural hypertension awareness and treatment were greater than those between urban and rural hypertension prevalence.

## DISCUSSION

Our study of 9859 adults in 22 diverse provinces in China demonstrated that urbanisation was associated with the prevalence, awareness and treatment of hypertension. The prevalence of hypertension in semiurbanised adults was lower than that in rural adults. The proportion of

**Table 1** Background characteristics of study participants aged 65 and over from Chinese Longitudinal Healthy Longevity Survey 2018 survey data (n=9859)

| Variables | Rural born | | | Urban born | Total | P value |
| | Non-urbanised | semiurbanised | Fully urbanised | | | |
|---|---|---|---|---|---|---|
| Prevalence, n (%) | | | | | | <0.001 |
| No | 1609 (38.89) | 1195 (41.90) | 522 (37.10) | 504 (34.45) | 3830 (38.84) | |
| Yes | 2528 (61.11) | 1657 (58.10) | 885 (62.90) | 959 (65.55) | 6029 (61.16) | |
| Awareness, n (%) | | | | | | <0.001 |
| No | 420 (21.07) | 255 (19.36) | 102 (13.08) | 69 (8.00) | 846 (17.08) | |
| Yes | 1573 (78.93) | 1062 (80.64) | 678 (86.92) | 794 (92.00) | 4107 (82.92) | |
| Treatment, n (%) | | | | | | <0.001 |
| No | 657 (33.17) | 418 (31.98) | 152 (19.56) | 97 (11.23) | 1324 (26.86) | |
| Yes | 1324 (66.83) | 889 (68.02) | 625 (80.44) | 767 (88.77) | 3605 (73.14) | |
| Age, n (%) | | | | | | <0.001 |
| Elderly age | 1045 (25.26) | 689 (24.16) | 257 (18.27) | 467 (31.92) | 2458 (24.93) | |
| Senile age | 1491 (36.04) | 1008 (35.34) | 546 (38.81) | 550 (37.59) | 3595 (36.46) | |
| Long livers | 1601 (38.70) | 1155 (40.50) | 604 (42.93) | 446 (30.49) | 3806 (38.60) | |
| Sex, n (%) | | | | | | <0.001 |
| Female | 2403 (58.09) | 1611 (56.49) | 708 (50.32) | 761 (52.02) | 5483 (55.61) | |
| Male | 1734 (41.91) | 1241 (43.51) | 699 (49.68) | 702 (47.98) | 4376 (44.39) | |
| Marital status, n (%) | | | | | | <0.001 |
| Not married | 2401 (58.04) | 1694 (59.40) | 793 (56.36) | 713 (48.74) | 5601 (56.81) | |
| Married | 1736 (41.96) | 1158 (40.60) | 614 (43.64) | 750 (51.26) | 4258 (43.19) | |
| Area, n (%) | | | | | | <0.001 |
| Eastern | 1867 (45.13) | 1198 (42.01) | 762 (54.16) | 1009 (68.97) | 4836 (49.05) | |
| Northeastern | 118 (2.85) | 56 (1.96) | 143 (10.16) | 121 (8.27) | 438 (4.44) | |
| Central | 1170 (28.28) | 715 (25.07) | 202 (14.36) | 136 (9.30) | 2223 (22.55) | |
| Western | 982 (23.74) | 883 (30.96) | 300 (21.32) | 197 (13.47) | 2362 (23.96) | |
| Education, mean (SD) | 2.23 (3.10) | 2.17 (3.12) | 5.09 (5.00) | 7.72 (5.21) | 3.44 (4.32) | <0.001 |
| Income, mean (SD) | 31 000.21 (32 620.93) | 32 741.37 (32 764.62) | 65 050.23 (33761.20) | 72 712.83 (31 715.16) | 42 553.08 (36 874.99) | <0.001 |
| BMI | | | | | | <0.001 |
| <24 | 2969 (71.77) | 2057 (72.12) | 862 (61.27) | 845 (57.76) | 6733 (68.29) | |
| ≥24 | 1168 (28.23) | 795 (27.88) | 545 (38.73) | 618 (42.24) | 3126 (31.71) | |
| Smoking, n (%) | | | | | | <0.001 |
| No | 3468 (83.83) | 2355 (82.57) | 1257 (89.34) | 1291 (88.24) | 8371 (84.91) | |
| Yes | 669 (16.17) | 497 (17.43) | 150 (10.66) | 172 (11.76) | 1488 (15.09) | |
| Drinking, n (%) | | | | | | 0.03 |
| No | 3508 (84.80) | 2420 (84.85) | 1224 (86.99) | 1277 (87.29) | 8429 (85.50) | |
| Yes | 629 (15.20) | 432 (15.15) | 183 (13.01) | 186 (12.71) | 1430 (14.50) | |
| Exercise regularly, n (%) | | | | | | <0.001 |
| No | 2911 (70.36) | 1896 (66.48) | 545 (38.73) | 551 (37.66) | 5903 (59.87) | |
| Yes | 1226 (29.64) | 956 (33.52) | 862 (61.27) | 912 (62.34) | 3956 (40.13) | |
| Medical insurance, n (%) | | | | | | |
| None | 342 (8.27) | 251 (8.80) | 260 (18.48) | 254 (17.36) | 1107 (11.23) | |
| UEBMI/URBMI | 249 (6.02) | 298 (10.45) | 829 (58.92) | 989 (67.60) | 2365 (23.99) | <0.001 |
| NCMS | 3472 (83.93) | 2255 (79.07) | 167 (11.87) | 87 (5.95) | 5981 (60.67) | |

Continued

**Table 1** Continued

| Variables | Rural born | | | Urban born | Total | P value |
|---|---|---|---|---|---|---|
| | Non-urbanised | semiurbanised | Fully urbanised | | | |
| Others | 74 (1.79) | 48 (8.80) | 151 (10.73) | 133 (9.09) | 406 (4.12) | |

Others: commercial medical insurance and government free medical insurance.
NCMS, new rural cooperative medical scheme insurance; UEBMI/URBMI, urban resident basic medical insurance /urban employee basic medical insurance.

awareness and treatment of hypertension among non-urbanised rural adults was lower than that among urban-born adults.

The prevalence of hypertension was lower in semiurbanised adults than in rural adults. This was highly relevant, given that a large proportion of floating people who migrated from rural to urban areas are young, healthy labourers,[27 28] including the younger-elderly population. Although some studies have examined the prevalence of hypertension in correlation with urbanisation,[15 29] they did not reveal the differences in demographic composition caused by urbanisation in detail. In contrast, our study was based on the sociodemographic characteristics of the floating population, suggesting that the proportion of young-elderly semiurbanised adults was large, so the prevalence of hypertension was low.

We also observed a significant association between the awareness and treatment of hypertension and years of education in urbanisation. Although some studies have documented a positive impact of education on the awareness and treatment of hypertension,[30 31] few have considered the impact of education in the context of urbanisation. We conclude that the higher the level of urbanisation, the higher the level of education. Better

**Table 2** Multivariate logistic regression model to determine factors associated with prevalence, awareness, treatment among participants aged 65 years and above

| Variables | Prevalence AOR (95% CI) | Awareness AOR (95% CI) | Treatment AOR (95% CI) |
|---|---|---|---|
| semiurbanised | 0.94 (0.90 to 0.99)* | 1.04 (0.95 to 1.14) | 1.02 (0.94 to 1.10) |
| Fully urbanised | 0.97 (0.92 to 1.03) | 1.04 (0.94 to 1.15) | 1.09 (1.00 to 1.19) |
| Urban born | 0.99 (0.94 to 1.03) | 1.10 (1.01 to 1.20)* | 1.17 (1.08 to 1.26)*** |
| Age: senile age | 1.12 (1.06 to 1.20)*** | 1.02 (0.91 to 1.14) | 1.08 (0.98 to 1.18) |
| Age: long livers | 0.98 (0.94 to 1.03) | 0.84 (0.77 to 0.91)*** | 0.85 (0.79 to 0.91)*** |
| Sex: male | 0.83 (0.75 to 0.91)*** | 0.81 (0.68 to 0.98)* | 0.74 (0.63 to 0.87)*** |
| Married | 0.91 (0.82 to 1.00) | 1.06 (0.88 to 1.29) | 1.12 (0.95 to 1.31) |
| Region: northeastern | 0.98 (0.88 to 1.09) | 0.81 (0.67 to 0.98)* | 0.84 (0.71 to 0.99)* |
| Region: central | 0.89 (0.86 to 0.92)*** | 0.98 (0.91 to 1.05) | 0.95 (0.90 to 1.01) |
| Region: western | 0.93 (0.91 to 0.96)*** | 0.90 (0.86 to 0.95)*** | 0.88 (0.85 to 0.92)*** |
| Education | 1.01 (0.99 to 1.02) | 1.03 (1.00 to 1.06)* | 1.04 (1.02 to 1.06)** |
| Income | 1.00 (1.00 to 1.00) | 1.00 (1.00 to 1.00)** | 1.00 (1.00 to 1.00)*** |
| BMI≥24 (kg/m) | 2.01 (1.83 to 2.22)*** | 1.43 (1.20 to 1.70)*** | 1.49 (1.28 to 1.72)*** |
| Smoking status: yes | 0.94 (0.83 to 1.07) | 1.05 (0.83 to 1.34) | 1.03 (0.853 to 1.27) |
| Alcohol use: yes | 0.97 (0.86 to 1.10) | 0.66 (0.53 to 0.83)*** | 0.71 (0.58 to 0.87)** |
| Exercise regularly: yes | 1.06 (0.97 to 1.17) | 1.01 (0.85 to 0.19) | 1.12 (0.97 to 1.30) |
| UEBMI/URBMI | 1.10 (0.94 to 1.29) | 1.67 (1.25 to 2.28)** | 1.40 (1.08 to 1.82)* |
| NCMS | 1.10 (0.95 to 1.27) | 1.03 (0.79 to 1.35) | 1.00 (0.79 to 1.26) |
| Others | 0.89 (0.70 to 1.14) | 1.24 (0.77 to 2.01) | 0.94 (0.62 to 1.41) |
| Constants | 1.43 (1.17 to 1.76)*** | 4.08 (2.80 to 5.92)*** | 2.03 (1.47 to 2.81)*** |

*p<0.05, **p<0.01, ***p<0.001.
Others: commercial medical insurance and government free medical insurance.
.AOR, adjusted OR; NCMS, new rural cooperative medical scheme insurance; UEBMI/URBMI, urban resident basic medical insurance /urban employee basic medical insurance.

**Table 3** Impact of urbanisation on the prevalence, awareness and treatment of hypertension stratified by different regions in China after controlling education and income

| Variables | Prevalence AOR (95% CI) | Awareness AOR (95% CI) | Treatment AOR (95% CI) |
|---|---|---|---|
| Eastern | 0.93 (0.88 to 0.99)* | 1.17 (1.04 to 1.32)** | 1.26 (1.14 to 1.40)*** |
| Northeastern | 0.97 (0.79 to 1.18) | 1.13 (0.79 to 1.63) | 1.00 (0.73 to 1.37) |
| Central | 1.05 (0.95 to 1.17) | 1.31 (1.05 to 1.63)* | 1.29 (1.08 to 1.55)** |
| Western | 1.11 (1.01 to 1.23)* | 1.28 (1.07 to 1.53)** | 1.34 (1.15 to 1.57)*** |

*p<0.05, **p<0.01, ***p<0.001.
AOR, adjusted OR.

education levels are often correlated with better family income. People with better educational levels may pay more attention to their health[32] and possibly have a higher healthcare expenditure budget. Therefore, improving the population education level, especially for rural adults, may have important practical and theoretical significance.

The prevalence of hypertension among older adults in eastern China was higher than that in the central and western China (table 2), which is similar to the results of previous studies.[16 33 34] The economic development and the level of urbanisation in eastern China is higher than those in central and western China.[11 35] Thus, the elderly living in eastern China may have high living standards, good living conditions, poor eating habits, long-term intake of high-salt and high-oil foods such as meat,[36] plus the lifestyle is fast-paced and high-pressured,[37] making them more likely to suffer from hypertension. Although the level of urbanisation in the western region has improved greatly, compared with the eastern region, there is a big gap in both the urbanisation rate and urban infrastructure construction.[11] Moreover, the urbanisation level of the provinces and regions within the western region is also very different, and this gap will continue to exist for some time. However, we also found that the higher the level of urbanisation within eastern China itself, the lower the prevalence of hypertension, indicating that urbanisation is a protective factor against hypertension. A possible reason for this is that eastern China is economically developed, and urbanisation has entered a normal state. Residents with a high level of urbanisation may focus on the prevention and care of chronic diseases, such as hypertension. Similar incidence characteristics of chronic diseases have been reported in developed countries, such as the USA,[38] reflecting the uneven distribution of medical resources between eastern and western China and between urban and rural areas.

The type of insurance a resident enrols in also has a different effect on hypertension awareness and treatment. Urban insurance enrolees had higher awareness and treatment than rural insurance enrolees and uninsured adults, which is consistent with earlier findings.[39] However, most floating people can only participate in the NCMS because of China's unique HRS (Hukou System). To receive full reimbursement, these floating people had to seek medical care in their hometowns.[40] Previous

**Table 4** Blinder-Oaxaca decomposition results between urban and rural adults

| | Prevalence | Awareness | Treatment |
|---|---|---|---|
| **Predicted probability** | | | |
| Urban | 61.18% (0.60% to 0.62%)*** | 85.61% (0.84% to 0.87%)*** | 77.37% (0.76% to 0.79%)*** |
| Rural | 61.11% (0.60% to 0.63%)*** | 78.93% (0.77% to 0.81%)*** | 66.83% (0.65% to 0.69%)*** |
| **Difference in predicted probability** | | | |
| Total difference | 0.01% (−0.02% to 0.02%) | 6.68% (0.04% to 0.09%)*** | 10.54% (0.08% to 0.13%)*** |
| Due to endowments (explained) | 2.31% (0.01% to 0.04%)** | 3.52% (0.01% to 0.06%)** | 6.56% (0.04% to 0.09%)*** |
| Due to coefficients (unexplained) | −0.02% (−0.04% to−0.00%)* | 2.82% (0.00% to 0.05%)* | 4.08% (0.01% to 0.07%)* |
| Due to interaction | 0.07% (−0.02% to 0.02%) | 0.00% (−0.02% to 0.03%) | −0.00% (−0.03% to 0.03%) |

*p<0.05, **p<0.01, ***p<0.001.

studies have shown that participation in local medical insurance for floating people is more conducive to the management and control of hypertension.[41 42] Therefore, further efforts are needed to integrate the health insurance system into a uniform system, which is beneficial for the management of hypertension and reduces obstacles in the process of urbanisation. We did not observe any association between commercial medical insurance and government-free medical insurance because there were only 406 other medical insurance cases documented in our study.

## Strengths and limitations

Our study has several strengths, including a large sample size from 500 urban and rural communities across 22 provinces in China, making it a useful supplement to the elderly group data for future studies. Further, the inclusion of multiple provinces and communities increases the generality of the results, and the standardisation and defining comprehensive and systematic information help to evaluate the results.

This study also has several limitations. First, we cannot make causal inferences since this study is cross-sectional by nature. However, the relationship between urbanisation and hypertension has been confirmed in some longitudinal studies,[16] and our sample is large and may be representative in China, and the research results may be credible. Second, this database is for people in China, and the results obtained can only be used in China and cannot be extended to other countries. However, it can provide useful insights for countries that are experiencing an ageing society, such as that of China.

## CONCLUSION

Urbanisation was significantly associated with the prevalence, awareness and treatment of hypertension, especially between non-urbanised rural adults and urban-born adults, in a representative national population in China. Our findings indicate the role of urbanisation that generates differences within demographic composition, education, regions and health insurance in shaping the distribution of prevalence, awareness and treatment of hypertension. Moreover, the results suggest that the relationship among hypertension prevalence, awareness, treatment and urbanisation varies within eastern, northeastern, central and western China and that public health programmes and policies for chronic diseases should be adjusted with the process of urbanisation and combined with individual-centred strategies.

**Acknowledgements** We thank the Chinese Longitudinal Healthy Longevity Survey, which provided the data in this research.

**Contributors** QY: conceptualisation, methodology, data analysis and writing. GZ: methodology, conceptualisation, writing with review, funding. All authors approved the final version of the paper. GZ is responsible for the overall content as guarantor.

**Funding** This study was supported by National Natural Science Foundation of China (71774102).

**Competing interests** None declared.

**Patient and public involvement** Patients and/or the public were not involved in the design, or conduct, or reporting, or dissemination plans of this research.

**Patient consent for publication** Not applicable.

**Ethics approval** This study involves human participants. The CLHLS study was approved by the Research Ethics Committee of Peking University (IRB00001052-13074), and all participants or their proxy respondents provided written informed consent. Participants gave informed consent to participate in the study before taking part.

**Provenance and peer review** Not commissioned; externally peer reviewed.

**Data availability statement** Data are available in a public, open access repository.

**ORCID iDs**
Qiutong Yu http://orcid.org/0000-0002-1758-0692
Genyong Zuo http://orcid.org/0000-0002-9305-9671

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
