## [Reviewer comments · BMJ Open]

ARTICLE DETAILS

TITLE (PROVISIONAL)	Impact of Urbanization on the Gaps of Hypertension Prevalence, Awareness, and Treatment among Older Age in China: A Cross-Sectional Study
AUTHORS	Yu, Qiotong; Zuo, Genyong

VERSION 1 – REVIEW

REVIEWER	ISIGUZO, Godsent University of Cape Town, Medicine
REVIEW RETURNED	07-Jan-2022

GENERAL COMMENTS	THE ROLE OF URBANIZATION IN IMPACTING ON THE URBAN-RURAL PREVALENCE, DIAGNOSIS AND MEDICATION OF HYPERTENSION GAPS IN OLDER AGE 1. Title and abstract : The term older adults look vague Suggestion: The role of urbanization in impacting on the urban-rural prevalence, diagnosis and medication of hypertension gaps in adults 65 years and above 2. Background/rationale: The background is good. However, I am not satisfied with the rationale for the study. The gap in knowledge was not explicit except the issue of 'floating population'. Previous studies which looked very similar were cited by the authors.
---

	There is also a 2021 similar work in China by Yu Q, Lin S and Wu J ‘Hypertension prevalence rates among urban and rural older adults of China, 1991-2015: a standardization and decomposition analysis’  3. Objectives: The objectives did not include the semi-urban population/floating population which was extensively highlighted as part of the rationale and reflected in the results and discussion 4. Study design: Good 5. Setting :Appropriate 6. Participant: Noted 7. Variables: Page 6; Line 33/34- medication was classified as yes if the respondents took anti-hypertensives – the sentence may be modified to reflect the period in which the anti-hypertensive was taken. 8. Data sources/measurement: Noted 9. Bias: It was a retrospective study and no special effort was shown to avoid bias as referred to in page 6 of checklist. 10. Study size: Noted 11. Quantitative variables: Noted 12. Statistical analysis: Noted 13. Participant: Noted 14. Descriptive data: Noted 15. Outcome data: Noted
--	--

16. **Main result:** Appropriate

17. **Other analysis:** Good

Discussion

18. **Key results:** The discussion on the difference between the prevalence of hypertension in semi-urbanized adult and rural adults was not elaborate.

The article looked at the floating population. However, the Blander-Oaxaca decomposition result interpretation was that more studies were needed for the floating population (line 35, page 13)

19. **Limitation:** Noted

Other comments

Page 4 line 27- Participation in China (Phrase not full sentence)

Page 4 line 31/32 Superscript 8 written in bigger character

Page 5 line 23/24 Superscript 8 written in bigger character

Page 6 line 58-59 The categorization of age by WHO 2015 was by Akhment Dyussenbayer (ref 18). The citation should be done properly; either the original WHO document is referenced or a secondary citation used.

Page 7 line 15-20: classification of BMI for Asian population was not very clear and not in keeping with cited article

Page 8 line 13/14 'This study will further use' future tense (work already done)

Page 17 number 3 reference list; line 7/8 and reference list

	number 28 page 18 Line 27/28- there is a space between the million and point. Conclusion There is need to rejig the grammar as pointed out in the pdf document, implement the modifications suggested above.
--	---

REVIEWER	Xing, Liying Liaoning Provincial Center for Disease Control and Prevention, Chronic Disease
REVIEW RETURNED	25-Jan-2022

GENERAL COMMENTS	The aim of this analysis was to examine the impact of urbanization on the prevalence, diagnosis, and medication of hypertension among elderly people in China. In general, I think that the authors have interesting data, but the manuscript need a lot of work to clarify. Firstly, the material methods and target population should be explained in detail. First of all, the sampling method and detailed investigation time node should be clarified. As shown in Figure 1, 16,000 people are eligible, and 15,874 people have completed the survey. However, it is said that 62 people were excluded because they were younger than 65 years old. According to the current understanding, these 62 people should be does not meet the inclusion criteria. 41 people were excluded due to technical reasons, and the specific reasons should be explained. 493 people refused to investigate, so these people should not be included in the 15,874 people who completed the investigation. 4549 people have missing data, please clarify which part of the data is missing. If the above exclusion reasons cannot be clearly stated, the representativeness of the data cannot be guaranteed. Secondly, the prevalence, diagnosis and treatment of hypertension should be further defined. According to the Chinese guidelines for the prevention and treatment of hypertension, hypertension is defined as systolic blood pressure greater than 140 mmHg and or diastolic blood pressure greater than 90 mmHg and taking antihypertensive drugs for the past two weeks. At present, the diagnosis of hypertension described in the article is actually the awareness rate of hypertension patients, and the treatment status is the medication rate. It is recommended to define the prevalence, awareness and treatment rates of hypertension in accordance with the latest guidelines. In the materials and methods, the blood pressure measurement method should be improved, which sphygmomanometer should be used, and how many times it was measured. Definitions of smoking, drinking, etc. should also be clarified. Thirdly, this paper is divided into eastern, central, western and northeastern regions according to regions. The prevalence of hypertension in different regions is quite different. It is recommended
---

	to stratify according to the fixed regions to analyze the impact of urbanization on the prevalence, awareness and treatment rates of hypertension. , which will provide meaningful data support for the formulation of hypertension prevention and control strategies in China. At the same time, the influence of education level and income level should be fully considered and analyzed, that is, the influence of different regions and different urbanization levels on the three rates of hypertension should be analyzed after fully adjusting education level and income level. Finally, the overall analysis idea of the full text should be redesigned according to the above comments.
--	---

VERSION 1 – AUTHOR RESPONSE

Response to Reviewer#1 comments:

1. Please confirm the approved referencing style and adhere to that

[Response]: We would like to thank Reviewer 1 for the time and effort in reviewing our manuscript and providing comments and suggestions, which have considerably helped us improve our manuscript. We have answered each of your points below and hope that our responses and revisions address all your comments.

We apologize for this mistake. The referencing style has been revised as per journal guidelines.

2. It will aid understanding to explain what this and other unconventional terms mean.

[Response]: Thanks for your constructive suggestion, we have revised “rural-to-urban migrants” to “floating people.” (page18, Line1, 3, 5)

3. Syntax error on page 5 “what extent the gaps in prevalence, awareness, and treatment of hypertension can be explained by urban-rural differences.”

[Response]: As per your comment, we have revised the sentences as follows (page 7 line 2): “the extent to which any gaps in prevalence, awareness, and treatment of hypertension can be explained by urban-rural differences.”

4. we defined hypertension as having a mean SBP \geq 140 mmHg or DBP \geq 90 mmHg or self-reported history of hypertension

[Response]: Many thanks for your suggestion. We have corrected the grammatical error and have changed the definition of hypertension based on the comments of Reviewer 2, as follows (page 8 line 5): “We defined hypertension as (1) systolic blood pressure (SBP) \geq 140 mmHg, diastolic blood pressure (DBP) \geq 90 mmHg, or (2) taking antihypertensive drugs.¹”

5. “The income of people born in cities was more than twice that of those born in rural areas.” Even for those who have migrated to the cities?

[Response]: According to your kind advice, in Table 1, the income of non-urbanized residents is 31,000 yuan, the income of semi-urbanized residents is 32,741 yuan, and the income of urban-born residents is 72,712 yuan. we revised the sentence as follows (page 12 line 17): “The income of people born in cities was more than twice that of those who were non-urbanized and semi-urbanized.”

6. “The association between geographical region and hypertension was significant difference.” check grammar

[Response]: We checked the grammar and revised the sentence as follows (page 12 line 16): “The association between geographical region and hypertension was significantly different.”

7. Married a

[Response]: We apologize for this inadvertent mistake. This spelling errors has been corrected.

8. We also observed a significant association between awareness, treatment of hypertension and the segment of education in urbanization, information not very clear.

[Response]: We sincerely apologize because this part was not clearly clarified in the manuscript. In Table 2, we found that for every additional year of education, hypertensive patients are 1.03 times more likely to know themselves to have hypertension, hypertensive patients are 1.04 times more likely to take medication. We revised the sentence as follows (page 16, line 11): “We also observed a significant association between the awareness and treatment of hypertension and years of education in urbanization.”

9. “Distinct insurance difference could observe in our research.” check grammar

[Response]: We apologize for this mistake. We have revised the sentence as follows (page 17 line 19): “The type of insurance a resident enrolls in also has a different effect on hypertension awareness and treatment.”

10. I don't think that this is a complimentary comment coming here!

[Response]: Thank you for underlining this deficiency. We have revised the Strengths and Limitation as follows (page 18 line 13): “Our study has several strengths, including a large sample size from 500 urban and rural communities across 22 provinces in China, making it a useful supplement to the elderly group data for future studies. Further, the inclusion of multiple provinces and communities increases the generality of the results, and the standardization and defining comprehensive and systematic information help to evaluate the results.

This study also has several limitations. First, we cannot make causal inferences since this study is cross-sectional by nature. However, the relationship between urbanization and hypertension has been confirmed in some longitudinal studies², and our sample is large and may be representative in China, and the research results may be credible. Second, this database is for people in China, and the results obtained can only be used in China and cannot be extended to other countries. However, it can provide useful insights for countries that are experiencing an aging society, such as that of China.”

11. Does this have any implication on the subject of the research?

[Response]: We appreciate your comment here that has helped us revise the Discussion. According to your suggestion, we revised the Strengths and Limitation as follows to prevent readers from ambiguity (page 18 line 18): “This study also has several limitations. First, we cannot make causal inferences since this study is cross-sectional by nature. However, the relationship between urbanization and hypertension has been confirmed in some longitudinal studies², and our sample is large and may be representative in China, and the research results may be credible. Second, this database is for people in China, and the results obtained can only be used in China and cannot be extended to other countries. However, it can provide useful insights for countries that are experiencing an aging society, such as that of China.”

12. But in line 15/16 page 26 above you said patient consent was not required

[Response]: We apologize for any confusion. Accordingly, we have added “Written informed consent was obtained from participants” in the Patient consent for publication section. We used the data from CLHLS. The Chinese Longitudinal Healthy Longevity Survey (CLHLS) was approved by the Ethical Review Committee of Peking University (IRB00001052–13074). All participants signed the informed consent at the time of participation. The research was performed in accordance with the Declaration

of Helsinki.

13. "PREFERENCES" check spelling

[Response]: We apologize for this inadvertent mistake. This spelling errors has been corrected.

Response to Reviewer#2 comments:

1. First, the material methods and target population should be explained in detail. First of all, the sampling method and detailed investigation time node should be clarified. As shown in Figure 1, 16,000 people are eligible, and 15,874 people have completed the survey. However, it is said that 62 people were excluded because they were younger than 65 years old. According to the current understanding, these 62 people should be does not meet the inclusion criteria. 41 people were excluded due to technical reasons, and the specific reasons should be explained. 493 people refused to investigate, so these people should not be included in the 15,874 people who completed the investigation. 4549 people have missing data, please clarify which part of the data is missing. If the above exclusion reasons cannot be clearly stated, the representativeness of the data cannot be guaranteed.

[Response]: We would like to thank Reviewer 2 for the time and effort in reviewing our manuscript and providing comments and suggestions, which have considerably helped us improve our manuscript. We have answered each of your points below and hope that our responses and revisions address all your comments.

In this study, we used secondary data from the 2018 CLHLS, which was collected in 2018. CLHLS using a multistage stratified proportional probability sampling design, approximately 16,000 elderly people in urban and rural communities were randomly selected from 500 sample areas in 22 provinces. We have revised the flowchart (Figure 1) to make the inclusion and exclusion criteria more detailed: 103 had technical problems, 62 were aged <65 years, 7 weighed <20 kg or >500 kg, 33 were < 1 meter tall, and 1 had >65 years of education. Furthermore, 4,549 people were excluded from the study. See Figure 1 for details of missing parts.

Figure 1 Study flowchart of participants selection (aged ≥ 65 years) from CLHLS 2018 survey data. Abbreviations: Chinese Longitudinal Healthy Longevity Survey (CLHLS)

2. Secondly, the prevalence, awareness and treatment of hypertension should be further defined. According to the Chinese guidelines for the prevention and treatment of hypertension, hypertension is defined as systolic blood pressure greater than 140 mmHg and or diastolic blood pressure greater than 90 mmHg and taking antihypertensive drugs for the past two weeks. At present, the awareness of hypertension described in the article is actually the awareness rate of hypertension patients, and the treatment status is the treatment rate. It is recommended to define the prevalence, awareness and treatment rates of hypertension in accordance with the latest guidelines.

In the materials and methods, the blood pressure measurement method should be improved, which sphygmomanometer should be used, and how many times it was measured. Definitions of smoking, drinking, etc. should also be clarified.

[Response]: The questions in the CLHLS about treatment for hypertensive patients are: "Whether to take antihypertensive medication." Definition of hypertension in the 2018 edition of the Guidelines for Prevention and Treatment of Hypertension is "whether to take medication within two weeks." There are some studies that used antihypertensive treatment to represent taking antihypertensive drugs for the past 2 weeks³ because the questionnaire only involves the issue of whether the respondents are taking antihypertensive drugs. Therefore, we used antihypertensive treatment to represent taking antihypertensive drugs for the past 2 weeks. We revised the definition of hypertension (page 8 line 5): "We defined hypertension as (1) systolic blood pressure (SBP) ≥ 140 mmHg, diastolic blood pressure (DBP) ≥ 90 mmHg, or (2) taking antihypertensive drugs.¹

After changing the definition of hypertension, there were 7 fewer people with hypertension and 7 more people with normal blood pressure

Hypertension awareness was defined as respondents knowing that they had been previously diagnosed with hypertension by a health professional, assessed by the question, "Have you been hospital diagnosed with hypertension?" (page 8, line 12)

Hypertension treatment was defined as taking blood pressure treatment, correlating with the question, "Whether you take antihypertensive drugs?" (page 8, line 14)

we added the blood pressure measurement method in the Variables of Discussion as follows (page 8, line 7): "After the study participant rested for at least 5 min, a research assistant measured the BP of the right arm twice using a mercury sphygmomanometer at 1-min intervals, and the mean of the two measurements was calculated. For bedridden participants, BP was measured in the reclining position." Manufacturer details of mercury sphygmomanometer including name or location were not mentioned in the CLHLS.

The definition of the variables was revised (page 10, line 2): "Participants who answered "smoking now" were recognized as yes, as was alcohol use and regular exercise."

3. Thirdly, this paper is divided into eastern, central, western and northeastern regions according to regions. The prevalence of hypertension in different regions is quite different. It is recommended to stratify according to the fixed regions to analyze the impact of urbanization on the prevalence, awareness and treatment rates of hypertension. , which will provide meaningful data support for the formulation of hypertension prevention and control strategies in China.

At the same time, the influence of education level and income level should be fully considered and analyzed, that is, the influence of different regions and different urbanization levels on the three rates of hypertension should be analyzed after fully adjusting education level and income level.

[Response]: Thanks for your constructive suggestion. After adjusting for years of education and income, we conducted a stratified analysis of urbanization and the prevalence, awareness, and treatment of hypertension in the eastern, central, western, and northeastern regions. For details, please refer to Table 3.

Table 3 Impact of urbanization on the prevalence, awareness and treatment of hypertension stratified by different regions in China after controlling education and income.

Variables	Prevalence	AOR (95%CI)	Awareness	AOR (95%CI)	Treatment	AOR (95%CI)
Eastern	0.93 (0.88, 0.99) *	1.17 (1.04, 1.32) **	1.26 (1.14, 1.40) ***			
Northeastern	0.97 (0.79, 1.18)	1.13 (0.79, 1.63)	1.00 (0.73, 1.37)			
Central	1.05 (0.95, 1.17)	1.31 (1.05, 1.63) *	1.29 (1.08, 1.55) **			
Western	1.11 (1.01, 1.23) *	1.28 (1.07, 1.53) **	1.34 (1.15, 1.57) ***			

CI, confidence interval; AOR, adjusted odds ratio. *** p < 0.001, ** p < 0.01, * p < 0.05

4. Finally, the overall analysis idea of the full text should be redesigned according to the above comments.

[Response]: Many thanks for your suggestion. First, in the third paragraph of Introduction, we added that the development of urbanization in eastern, central, western, and northeastern China is different and that urbanization in different regions of China also has differences in the prevalence of hypertension (page 5, line 1): "Due to the imbalance between urban and rural development as well as

the different levels of urbanization development in eastern, northeastern, central, and western China,⁴ there are gaps in the awareness and treatment of hypertension.⁵

In the fifth paragraph of the Introduction, we conclude from the references, there have been few studies on the awareness and treatment of urbanization and hypertension in the context of different economic regions throughout China (page 5, line 17): “Furthermore, there is a strong body of evidence for an increased risk of hypertension in children and adolescents in eastern, central, and western China, showing differing socioeconomic profiles.⁶”

In the last paragraph of the Introduction, we present the purpose of the study, whether the level of urbanization was associated with the prevalence, awareness, and treatment of hypertension in eastern, northeastern, central, and western China after controlling for education and income (page 6, line 18): “The main purpose of this study was to examine (1) whether the level of urbanization is associated with the prevalence, awareness, and treatment of hypertension in eastern, northeastern, central, and western China after controlling for education and income and (2) whether there is a gap in prevalence, awareness, and treatment of hypertension between urban-born and non-urbanized rural adults, and (3) the extent to which any gaps in prevalence, awareness, and treatment of hypertension can be explained by urban-rural differences.”

Second, we have added the method of stratified analysis to the Statistical Analysis section as follows (page 11, line 3): “In addition, subgroup analyses were performed using a logistic model to examine the impact of urbanization on the prevalence, awareness, and treatment of hypertension stratified by region in China after controlling for education and income.

Third, we included the results on urbanization and the prevalence, awareness, and treatment of hypertension in eastern, central, western, and northeastern China in the Results, as follows (page 14, line 6): “Table 3 lists the results of the stratified analyses. Regarding geographic location, the prevalence of hypertension decreased by a factor of 0.93 for each level of urbanization in eastern China (OR=0.93, 95% CI=0.88 to 0.99; P<0.05). Additionally, the relationship between urbanization and the prevalence of hypertension was significant in western China, and participants were 1.11 times more likely to develop hypertension for each level of urbanization (OR=1.11, 95% CI=1.01 to 1.23; P<0.05). No significant association was noted between urbanization and the prevalence of hypertension in northeastern or central China. The urbanization level was associated with hypertension awareness and treatment in eastern (OR=1.17, 95% CI=1.04 to 1.32; P<0.01; OR=1.26, 95% CI=1.14 to 1.40; P<0.001), central (OR=1.31, 95% CI=1.05 to 1.63; P<0.05; OR=1.29, 95% CI=1.08 to 1.55; P<0.01), and western (OR=1.28, 95% CI=1.07 to 1.53; P<0.01; OR=1.34, 95% CI=1.15 to 1.57; P<0.001) China. No significant associations were noted between urbanization and hypertension awareness and treatment in northeast China.

Fourth, we revised the paragraph in the Discussion on the relationship between the prevalence, awareness, and treatment of hypertension and urbanization in the eastern, central, western and northeast China. The revised paragraph was as follows (page 16, line 20): “The prevalence of hypertension among older adults in eastern China was higher than that in the central and western China (Table 2), which is similar to the results of previous studies.^{2 7 8} The economic development and the level of urbanization in eastern China is higher than those in central and western China.^{9 5} Thus, the elderly living in eastern China may have high living standards, good living conditions, poor eating habits, long-term intake of high-salt and high-oil foods such as meat,¹⁰ plus the lifestyle is fast-paced and high-pressured,¹¹ making them more likely to suffer from hypertension. Although the level of urbanization in the western region has improved greatly, compared with the eastern region, there is a big gap in both the urbanization rate and urban infrastructure construction.⁵ Moreover, the urbanization level of the provinces and regions within the western region is also very different, and this gap will continue to exist for some time. However, we also found that the higher the level of

urbanization within eastern China itself, the lower the prevalence of hypertension, indicating that urbanization is a protective factor against hypertension. A possible reason for this is that eastern China is economically developed, and urbanization has entered a normal state. Residents with a high level of urbanization may focus on the prevention and care of chronic diseases, such as hypertension. Similar incidence characteristics of chronic diseases have been reported in developed countries, such as the United States,12 reflecting the uneven distribution of medical resources between eastern and western China and between urban and rural areas.”

Finally, we included statistical results on the prevalence, awareness, and treatment of urbanization and hypertension in the eastern, central, western, and northeastern regions in the abstract results, as follows (page 2 line 16): “Urbanization in eastern (OR=0.93, 95% CI=0.88 to 0.99; P<0.05) and western (OR=1.11, 95% CI=1.01 to 1.23; P<0.05) China was associated with the prevalence of hypertension. The urbanization level was also associated with hypertension awareness and treatment in eastern (OR=1.17, 95% CI=1.04 to 1.32; P<0.01; OR=1.26, 95% CI=1.14 to 1.40; P<0.001), central (OR=1.31, 95% CI=1.05 to 1.63; P<0.05; OR=1.29, 95% CI=1.08 to 1.55; P<0.01), and western (OR=1.28, 95% CI=1.07 to 1.53; P<0.01; OR=1.34, 95% CI=1.15 to 1.57; P<0.001) China.”

References

- 1 Organization WH. A Global Brief on Hypertension: Silent Killer, Global Public Health Crisis, 2013.
- 2 Li J, Shi L, Li S, et al. Urban-rural disparities in hypertension prevalence, detection, and medication use among Chinese Adults from 1993 to 2011. *Int J Equity Health* 2017;16(1):50.
- 3 Yao Y, Cao K, Zhang K, et al. Residential Proximity to Major Roadways and Prevalent Hypertension Among Older Women and Men: Results From the Chinese Longitudinal Healthy Longevity Survey. *Front Cardiovasc Med* 2020;7:587222.
- 4 Wei H, Li L, Nian M. China’s Urbanization Strategy and Policy During the 14th Five-Year Plan Period. *Chinese Journal of Urban and Environmental Studies* 2021;09(01)
- 5 Maimaitiming A, Xiaolei Z, Huhua C. Urbanization in Western China. *Chinese Journal of Population Resources and Environment* 2013;11(1):79-86.
- 6 Fan Z, Liao Z, Zong X, et al. Differences in prevalence of prehypertension and hypertension in children and adolescents in the eastern, central and western regions of China from 1991-2011 and the associated risk factors. *PLoS One* 2019;14(1):e0210591.
- 7 Lu J, Lu Y, Wang X, et al. Prevalence, awareness, treatment, and control of hypertension in China: data from 1.7 million adults in a population-based screening study (China PEACE Million Persons Project). *The Lancet* 2017;390(10112):2549-58.
- 8 Kristi R DG, Paul M, Xigui W, Jing C, Guanyong H, Xiufang D, Paul K. Wheltona, and Jiang H. Geographic variations in the prevalence, awareness, treatment and control of hypertension in China. *Journal of Hypertension* 2003;21:1273-80.
- 9 Yu X, Zhang W. All-cause mortality rate in China: do residents in economically developed regions have better health? *Int J Equity Health* 2020;19(1):12.
- 10 Wen F, Sun R, Ziaei SM. Research on the relationship between the imbalance of regional economic growth and the allocation of financial resources. *E3S Web of Conferences* 2021;275
- 11 Tin Tin Su1 HAM, Azmi Mohamed Nahar, Nurul Ain A. The effectiveness of a life style modification and peer support home blood pressure monitoring in control of hypertension: protocol for a cluster randomized controlled trial. *BMC Public Health* 2014;14:1-7.
- 12 Ostchega Y, Hughes JP, Zhang G, et al. Differences in Hypertension Prevalence and Hypertension Control by Urbanization Among Adults in the United States, 2013-2018. *Am J Hypertens* 2022;35(1):31-41.

VERSION 2 – REVIEW

REVIEWER	Xing, Liying Liaoning Provincial Center for Disease Control and Prevention, Chronic Disease
REVIEW RETURNED	10-Mar-2022
GENERAL COMMENTS	Thanks to the author's work, the overall revision of the article has met expectations. It will provide certain data support for the formulation of hypertension prevention and control strategies in different regions of China. It also provides a reference for the formulation of differentiated prevention and control strategies for hypertension in different regions of the world.